# Research on Performance Degradation Estimation of Key Components of High-Speed Train Bogie Based on Multi-Task Learning

**DOI:** 10.3390/e25040696

**Published:** 2023-04-20

**Authors:** Junxiao Ren, Weidong Jin, Yunpu Wu, Zhang Sun, Liang Li

**Affiliations:** 1School of Electrical Engineering, Southwest Jiaotong University, 999 Xi’an Road, Chengdu 611756, China; 2China-ASEAN International Joint Laboratory of Integrated Transportation, Nanning University, 8 Longting Road, Nanning 541699, China; 3School of Electrical Engineering and Electronic Information, Xihua University, 9999 Hongguang Road, Pidu District, Chengdu 610097, China

**Keywords:** high-speed train, signal processing, fault diagnosis, performance degradation, convolution neural network, multi-task learning

## Abstract

The safe and comfortable operation of high-speed trains has attracted extensive attention. With the operation of the train, the performance of high-speed train bogie components inevitably degrades and eventually leads to failures. At present, it is a common method to achieve performance degradation estimation of bogie components by processing high-speed train vibration signals and analyzing the information contained in the signals. In the face of complex signals, the usage of information theory, such as information entropy, to achieve performance degradation estimations is not satisfactory, and recent studies have more often used deep learning methods instead of traditional methods, such as information theory or signal processing, to obtain higher estimation accuracy. However, current research is more focused on the estimation for a certain component of the bogie and does not consider the bogie as a whole system to accomplish the performance degradation estimation task for several key components at the same time. In this paper, based on soft parameter sharing multi-task deep learning, a multi-task and multi-scale convolutional neural network is proposed to realize performance degradation state estimations of key components of a high-speed train bogie. Firstly, the structure takes into account the multi-scale characteristics of high-speed train vibration signals and uses a multi-scale convolution structure to better extract the key features of the signal. Secondly, considering that the vibration signal of high-speed trains contains the information of all components, the soft parameter sharing method is adopted to realize feature sharing in the depth structure and improve the utilization of information. The effectiveness and superiority of the structure proposed by the experiment is a feasible scheme for improving the performance degradation estimation of a high-speed train bogie.

## 1. Introduction

With the rapid development of high-speed trains, the safety, stability, and comfort of trains have attracted much attention. The high-speed train bogie is a key component connecting the track and the high-speed train’s body, the structure of which is shown in Figure 1. The bogie can absorb and cushion the impact and vibration from the wheel–rail contact, making the vehicle body more stable during operation. At the same time, the bogie can depress the serpentine movement of the train, prevent the huge impact between the wheel and the rail, and reduce the risk of train derailment [1]. The health condition of the bogie is directly related to the safe and stable operation of the train. However, during the long-term service of the train, the bogie is subject to vibration and impact caused by wheel–rail contact, and the damper components on the bogie inevitably suffer from performance degradation and failure. Therefore, detecting the health condition of the bogie to maximize its operating condition at all times is an important means to ensure the safe operation of the train [2].

At present, bogie health condition detection mainly includes fault diagnosis and performance degradation estimation [3]. Due to the complex structure of the bogie and the irregularity of wheel–rail contact, physical modeling of a train’s operation is difficult to complete [4]. Therefore, current studies are usually carried out in a data-driven manner. The high-speed train vibration signal is a nonlinear and complex signal [1], and the metrics obtained by manual feature extraction methods, such as information entropy, cannot accurately achieve performance degradation estimations [2]. Deep learning algorithms can avoid the process of manual feature selection and learn the key information contained in the signal more effectively and adaptively, which is a promising alternative to the traditional information theory and signal processing methods for processing complex signals and has achieved a large number of valuable research results.

In the research on fault diagnosis of high-speed train bogies, Huang et al. [5] proposed a novel bogie fault diagnosis method for high-speed trains by improving the empirical modal decomposition algorithm and combining it with a one-dimensional convolutional neural network. The method employs the IMF component of the vibration signal as the model input to extract fault features more accurately and obtained 99.3% and 98.7% estimation accuracy on two bogie data sets. Chen et al. [6] proposed a fault diagnosis model combined with CapsNet to complete fault identification and classification under seven operating conditions; the recognition accuracy reached 96.65%, which proved the potential of CapsNet in high-speed train fault diagnosis. Wu et al. [7] proposed a Bayesian deep learning-based unexpected fault detection method for high-speed train bogies to identify between known and unknown faults. Qin et al. [8] proposed a stepwise adaptive convolutional network to achieve bogie fault diagnosis of high-speed trains at different speeds with a 96.1% recognition accuracy. These studies have completed fault diagnoses of different damper components of the bogie, which makes it possible to diagnose the fault status of all dampers at the same time. However, the disadvantage is that it is impossible to judge the fault degree of the damper, that is, the performance status of each damper.

In the research on performance degradation estimations of high-speed train bogies, Ren et al. [2] proposed a novel grouping-attention convolutional neural network to implement performance degradation estimations of lateral dampers for high-speed trains. The model fully considers the differences and similarities between different channels of vibration signals, and the signals of different channels are grouped as the input of the model, which improves the efficiency of key feature learning and greatly reduces the error of performance degradation estimations. Qin et al. [3] proposed a novel multiple convolutional recurrent neural network that implements the performance degradation estimation task of bogie-related critical damper components by classification. Ren et al. [9] combined LSTM and CNN to propose a novel 1D-ConvLSTM time-distributed convolutional neural network, which is a structure that learns performance degradation trend features and achieves the estimation of unknown performance degradation states. These studies can be used to estimate the failure degree of the damper, that is, the performance degradation state. The advantage of these studies is that it is possible to accurately obtain the current state of the damper and formulate a more accurate maintenance plan. However, the disadvantage is that the performance degradation state estimation can only be completed for one damper alone and cannot be performed for all dampers at the same time. It takes substantial computing resources to complete the performance degradation state estimation of all dampers.

The deep multi-task learning model can achieve different objectives of tasks at the same time [10]. In theory, it can simultaneously estimate the performance degradation of different bogie dampers [11]. In terms of data, the vibration signal of a high-speed train is adopted as the experimental data in the current research and the research of this paper. The data are obtained by multiple sensors on the bogie and the train body simultaneously collecting the operational vibration state of the train [4,12,13]. In other words, the vibration signal of the high-speed train contains the information of all dampers, which provides a solid data base for multi-task learning. Based on the deep multi-task learning model, this paper proposes a multi-task and multi-scale convolutional neural network (MMCNN). The proposed model takes into account the advantages of fault diagnosis research and performance degradation estimation research and can simultaneously estimate the performance degradation state of all dampers on the bogie. The proposed model is based on the improvement of the Google multi-gate mixture-of-experts (MMoE) [14] and adopts the multi-task learning model paradigm of soft parameter sharing, fully taking into account the relationship and difference between different dampers. At the same time, considering the complex frequency components and multi-scale characteristics of high-speed train vibration signals [15], a large number of multi-scale convolution modules are adopted in the proposed model to replace the traditional convolution layers in the original MMoE model, which better realizes multi-scale feature learning. In the proposed model, the depth gate structure is adopted to distribute the weight of the output characteristics of different shared branches, so as to better realize the accurate estimation of the performance degradation state of different dampers. In general, the innovations in this paper are as follows:Based on Google MMoE, this paper proposes a multi-task and multi-scale convolutional neural network (MMCNN), which is a multi-task learning model that can simultaneously estimate the performance degradation of multiple dampers. The proposed model can realize the monitoring of fault degree (i.e., performance degradation state) on the basis of the existing research on bogie fault diagnosis of high-speed trains. In addition, the model can also estimate multiple dampers simultaneously on the basis of the existing research on performance degradation estimations, which expands the means of health status monitoring of high-speed trains.In view of the multi-scale characteristics of high-speed train vibration signals, the proposed model applies a multi-scale one-dimensional convolution module instead of an ordinary convolution layer on the basis of MMoE to better realize multi-scale feature learning.Taking into account the multi-channel characteristics of high-speed train vibration signals, a new gate structure, which is an attention mechanism considering global and local relations, is adopted in the proposed model. It can more accurately measure the importance of the features output by different feature branches for the performance degradation state estimation of different dampers to achieve more reasonable feature allocation.

## 2. Data Description

The monitoring data of high-speed trains can be obtained from trains in service and also from the rolling and vibration test rigs for vehicles. Although the data obtained through the above methods can more truly reflect the vibration mode of high-speed trains, the methods require a high amount of human and material resources. At the same time, some data under extreme performance degradation states cannot be obtained from the rolling and vibration rig for vehicles or trains in service, which seriously threatens the safety of the train operation and rolling test rig [16]. Therefore, for safety reasons, the vibration signals of high-speed trains adopted in this paper mainly come from simulation experiments, which are also the commonly applied means in this field [17]. The high-speed train simulation model in the simulation experiment is based on the rolling and vibration test rig for vehicles developed by the State Key Laboratory of Traction Power of Southwest Jiaotong University, which is the world’s most advanced rolling and vibration combined test rig with independent excitations of left and right rollers. Figure 2 shows photos of the rolling and vibration test rig for vehicles and the tested vehicle (CRH380A). The simulation experiment adopts the operating environment parameter settings and the distributed structure of the sensor network, which are completely consistent with the rolling rig, and adopts the measured track excitation. By changing the relevant structural parameters, the monitoring data information that is difficult to generate by the rolling and vibration tests and the trains in service is provided, which meets the need for obtaining the experimental data under extreme circumstances. It is worth noting that, in order to approach the real operation state as much as possible, the measured track excitation spectrum of the Wuhan–Guangzhou line was applied as the simulation track excitation in the simulation.

The simulation experiment set 58 monitoring points in the CHR380A simulation model, recorded the vibration of the train and of the key positions on the bogie, and obtained a total of 58 channels (including 29 acceleration signals and 29 displacement signals) of high-speed train vibration signal data. The type, quantity, location, and sampling frequency (243 Hz) of these 58 monitoring points were all set according to the standards of the rolling and vibration test rig for vehicles in the State Key Laboratory of Traction Power of Southwest Jiaotong University. The locations of the 58 monitoring points are shown in Figure 3. The details of the collected high-speed train monitoring data channels are shown in Table 1. Figure 4 shows some high-speed train vibration signal samples. Figure 5 shows the vibration signal samples of a high-speed train lateral damper (Lat_1) in different performance degradation states at the same driving position and time interval. The vibration modes of these samples are very close and it is not easy to accurately estimate their performance degradation.

## 3. Methods

The MMCNN network proposed in this paper is a multi-task learning model based on multi-gate mixture-of-experts (MMoE), which applies soft parameter sharing. Compared with the baseline method (such as shared-bottom structure [10]) which widely applies the hard parameter sharing, MMoE has a better effect when the task correlation is low. This is because the hard parameter-sharing multi-task learning model does not always outperform the corresponding single-task model in all tasks. Many DNN-based hard parameter-sharing multi-task learning models are sensitive to factors such as data distribution differences and the relationship between tasks [10]. The internal conflicts caused by task differences will actually damage the prediction of at least some tasks, especially when the parameters of the models are widely shared among all tasks [14,18].

It is obviously a multi-task learning problem to realize the performance degradation estimation of multiple high-speed train dampers at the same time. Different types of dampers (such as lateral dampers and yaw dampers) and different installation positions of the same type of dampers (installed on different bogies at the front and rear) will make the correlation between the estimated tasks of different dampers low. This type of task has great differences in feature representation in the shared representation layer. The effect of the shared representation layer is not so obvious and it is more likely to have conflicts or noise, which is closer to the single-task model. Therefore, this paper combines the characteristics of performance degradation estimation tasks and multi-task learning of high-speed train bogie dampers and adopts the multi-task learning model structure based on soft parameter sharing such as MMoE. The proposed model can not only handle tasks with high correlations, but also obtain better results with low correlations.

MMoE can explicitly model the relationship between multiple tasks (which can be understood as learning different aspects of all tasks), the specific structure of which is shown in Figure 6. MMoE can learn the characteristics of different tasks through a gate structure (a kind of deep attention mechanism) and assign the specific weight of different aspects of tasks on this basis. Compared with the shared-bottom multi-task model, MMoE can improve the performance of the model by capturing the differences between different tasks without significantly adding more model parameters.

A high-speed train vibration signal is a nonlinear and non-stationary multi-channel signal with complex frequency components and multi-scale characteristics [1]. Considering the multi-scale characteristics of a high-speed train vibration signal, a multi-task and multi-scale convolutional neural network (MMCNN) is proposed in this paper, the overall structure of which is shown in Figure 7. The structure consists of four parts: the multi-scale shared representation part, the gate structure, the high-dimensional feature learning part, and the output part. Compared with the original MMoE structure, a one-dimensional multi-scale convolution neural module is adopted in these four parts to replace the ordinary convolution layer to achieve better multi-scale feature learning and make the estimation results more accurate.

### 3.1. Multi-Scale Shared Representation Part

The multi-scale shared representation part is mainly composed of three identical convolutional networks. Each convolutional network is composed of a one-dimensional convolution layer and three one-dimensional multi-scale inception convolution modules (1D-IC-Module) in series. The specific structure of this part is shown in Figure 8. This part contains three branches with the same structure, and the specific hyperparameters of one branch are shown in Table 2. The 1D-IC-Module is improved based on Inception V1 [19]. As Inception V1 is developed to process two-dimensional image data and the vibration signal processed in this paper is one-dimensional signal data, this paper makes corresponding improvements on the basis of the original inception. The 1D-IC-Module is the modified structure, and its specific structure and hyperparameters are shown in Figure 9, where PW-CNN represents point convolution and 1D-CNN represents one-dimensional convolution with a convolution kernel length of three. The multi-scale shared representation part composed of 1D-IC-Module can obtain the feature expression of a high-speed train vibration signal at different scales and improve the multi-scale feature learning ability of the model.

### 3.2. Gate Structure

The gate structure is a kind of attention mechanism based on SE-Net [20], and on the basis of the SE-Net structure, a local correlation calculation is added. The proposed gate structure can better judge the relationship between different tasks by combining the global and local relevance of the tasks. Its specific structure hyperparameters are shown in Figure 10, where GAP represents the global average pooling layer and FC represents the fully connected layer.

### 3.3. High-Dimensional Feature Learning Part

The high-dimensional feature learning part is composed of four one-dimensional depthwise separable convolution modules (1D-DSC-Modules), whose specific structure is shown in Figure 11 and whose specific hyperparameters are shown in Table 3. The 1D-DSC-Module consists of three one-dimensional depthwise separable convolutions (1D-DSCNN) [21] and a pointwise convolution (PW-CNN). This design of the 1D-DSC-Module referred to existing research results of high-speed train bogie performance degradation estimations [2,9]. The specific structure and hyperparameters of the 1D-DSC-Module are shown in Figure 12; the module has low computational complexity and multi-scale feature learning capability. Combining the capabilities of the 1D-DSC-Modules, the high-dimensional feature learning part can further learn the features from the multi-scale shared representation part while reducing the computational load and the complexity of the model.

### 3.4. Output Part

The output part first applies the 1D-DS-Module to adjust the dimension of input features, and then adopts the global average pooling layer to compress and tile features to avoid damaging the time-scale characteristics of features in the process of feature flattening [22], and finally, obtains the performance degradation estimation results through two fully connected layers with Relu. The specific structure of the output part is shown in Figure 13, and the specific hyperparameters are shown in Table 4.

## 4. Experiments

In this paper, a multi-task and multi-scale convolutional neural network (MMCNN) was proposed that can simultaneously estimate the performance degradation of multiple high-speed train bogie dampers. The experiment part first carried out ablation experiments and tested the multi-scale convolution module and gate structure applied to the structure to verify the effectiveness of these improvements. Then, another experiment was carried out to test the difference between the performance degradation estimation for all sensors and for only the same type of damper, considering the range of multiple tasks. Finally, the proposed MMCNN was compared with other multi-task learning structures to verify its effectiveness and superiority.

The sample size of the high-speed train vibration signal adopted in the experiment is 243 × 58. Here, 243 refers to the sampling points included in 1s under the sampling frequency of 243 Hz, and 58 refers to the total number of channels of the high-speed train’s vibration signal. A train compartment consists of four lateral dampers and eight yaw dampers, among which two yaw dampers are redundant in design, that is to say, the yaw dampers can be simplified into four groups. The four lateral dampers are labeled Lat_1, Lat_2, Lat_3, and Lat_4, and the four yaw damper groups are labeled Yaw_1, Yaw_2, Yaw_3, and Yaw_4. The training set of mixed performance degradation data of all lateral dampers and yaw dampers is shown in Table 5, and the test set is shown in Table 6. The training set of performance degradation data of the four lateral dampers is shown in Table 7, and the test set is shown in Table 8. The training set of performance degradation data of four groups of yaw dampers is shown in Table 9, and the test set is shown in Table 10. It is worth noting that 20% of the samples in the training set were randomly selected as the validation set during the training process.

The evaluation indicators of the model are MAE (1) and RMSE (2).
(1)$$MAE\left(\hat{y},y\right)=\frac{1}{n}\sum_{i=1}^{n}\left|{\hat{y}}_{i}-y_{i}\right|$$
(2)$$RMSE\left(\hat{y},y\right)=\sqrt{\frac{1}{n}\sum_{i=1}^{n}{\left({\hat{y}}_{i}-y_{i}\right)}^{2}}$$
where $\hat{y}$ represents the estimation result, $y_{i}$ represents the true label, and *n* represents the number of $\hat{y}$.

RMSE can better reflect whether there are too many large errors. MAE reflects the evaluation of errors. All experiments were carried out with Python (using Keras, TensorFlow) on a PC with 2.80 GHz × 4 CPU and 32 GB memory. An NVIDIA 1080Ti GPU card was used for acceleration.

### 4.1. Ablation Experiments

The MMCNN proposed in this paper applies the 1D-IC-Module and 1D-DSC-Module multi-scale convolution modules in the multi-scale shared representation part and high-dimensional feature learning part, respectively, to better realize multi-scale feature learning. An ablation experiment was carried out for the improvement of the multi-scale convolution module, adopting the experimental data in Table 5 and Table 6. Firstly, the performance degradation estimation results of applying the common convolution layer and other different multi-scale convolution modules in the multi-scale shared representation part were compared, and the results are shown in Table 11. Secondly, the performance degradation estimation results of applying an ordinary convolution layer and other multi-scale convolution modules in the high-dimensional feature learning part were compared, and the results are shown in Table 12. It can be seen from the estimation results that the error of the estimation results obtained by applying the convolution module with multi-scale learning ability is smaller than that obtained by applying the ordinary convolution layer. At the same time, among these convolution modules with multi-scale learning ability, the error of performance degradation estimation results obtained by applying the 1D-IC-Module and 1D-DSC-Module proposed in this paper is the minimum.

The ablation experiment also discussed the impacts on performance degradation estimation results while adopting different amounts of 1D-IC Modules and 1D-DSC-Modules. Firstly, the experiment compared the situations of using different quantities of 1D-IC Modules in the multi-scale shared presentation part and the experimental results are shown in Table 13. The experimental results indicate that the minimum estimation error can be achieved when three 1D-IC modules are applied in the multi-scale shared representation part. Secondly, the experiment compared the situations of adopting different numbers of 1D-DSC-Modules in the high-dimensional feature learning part and the experimental results are shown in Table 14. The experimental results indicate that the minimum estimation error can be achieved when four 1D-DSC-Modules are applied in the high dimensional feature learning part. The experimental results also demonstrate that adopting different amounts of multi-scale modules can have significant impacts on the estimation results. When the proposed structure is applied to performance degradation estimation tasks in other fields, the number of multi-scale modules adopted needs to be adjusted based on actual experimental results.

The gate structure proposed in this paper considers not only the global correlation between channels but also the local correlation between different channels, so as to better share the features of different tasks. Another ablation experiment is carried out to compare the impact of the original SE-Net and of the improved gate structure on the estimation of multi-task performance degradation. The experimental results are shown in Table 15. It can be seen from the results that, after considering local correlation, the feature allocation between tasks can be achieved more accurately, and the accuracy of performance degradation estimations can be improved.

### 4.2. Multi-Task Range Experiments

This experiment tested the three data sets in Table 5 with Table 6, Table 7 with Table 8, and Table 9 with Table 10 to explore the impacts of different dampers on the estimation for multi-task performance degradation. The experimental results are shown in Table 16. It can be seen from the results that the error is obviously low when the multi-task performance degradation estimation is conducted only for the lateral damper and the yaw damper, while the error is high when the multi-task performance degradation estimation is conducted for all dampers. That is to say, the task correlation of performance degradation estimation for different dampers is low, and the task correlation of performance degradation estimation of the same type of dampers is high. The experimental results can be utilized as a reference for practical problems, and the range of multi-task coverage can be considered according to actual conditions and needs.

### 4.3. Comparison Experiments

In order to verify the effectiveness and the superiority of the proposed MMCNN, this paper adopted other different multi-task learning models for comparison experiments, including the tasks-constrained deep convolutional network (TCDCN) [23], multi-task network cascades (MNCs) [24], multi-gate mixture-of-experts (MMoE) [14], SubNetwork Routing (SNR) [25], and progressive layered extraction (PLE) [26]. Other multi-task learning models apply one-dimensional convolution instead of the two-dimensional convolution in the original model, so as to ensure the same experimental environment. The experimental data adopt the mixed performance degradation data in Table 5 and Table 6, and the experimental results are shown in Table 17. It can be seen from the result that, compared with the hard parameter sharing multi-task learning model, the soft parameter sharing multi-task learning model performs better, which, once again, verifies that the correlation between the performance degradation estimation tasks of different dampers is low. In the soft parameter sharing multi-task learning model, the MMCNN proposed in this paper obtained the minimum estimation error and has obvious advantages. Figure 14 shows the process of loss decline during the first 200 epochs of training. It can be seen that the convergence speed and stability of the proposed MMCNN are optimal.

## 5. Conclusions

In this paper, a multi-task and multi-scale convolutional neural network (MMCNN) was proposed to estimate the performance degradation of multiple high-speed train bogie dampers. Compared with the models in the existing fault diagnosis research, this model can realize the quantitative analysis of different dampers. Compared with the models in the existing performance degradation studies, the proposed model can estimate the performance degradation of multiple dampers at the same time. The proposed model is a soft parameter sharing multi-task learning model that is improved based on the Google MMoE model combined with the multi-scale characteristics of a high-speed train vibration signal. In the model, 1D-IC-Module and 1D-DSC-Module multi-scale convolution modules are applied to realize multi-scale task learning of signals. At the same time, considering the correlation between different dampers, an attention mechanism gate structure that can calculate the global and local correlation is proposed to complete the allocation of shared features in the multi-task model. Compared with other excellent multi-task models, the proposed MMCNN achieves the minimum estimation error and proves its effectiveness and superiority.

## Figures and Tables

**Figure 1 entropy-25-00696-f001:**
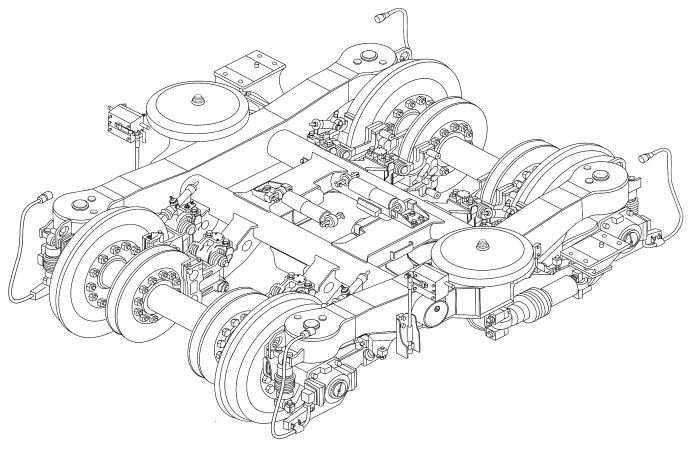
Overall structure of high-speed train bogie.

**Figure 2 entropy-25-00696-f002:**
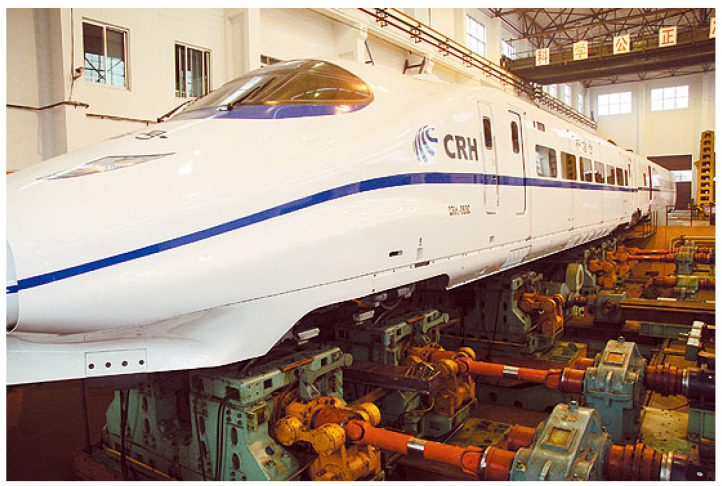
Rolling and vibration test rig for vehicle and the tested vehicle (CRH380A).

**Figure 3 entropy-25-00696-f003:**
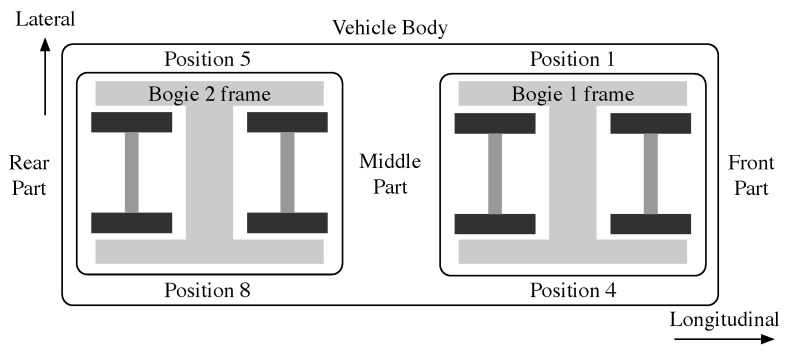
The locations of the 58 monitoring points.

**Figure 4 entropy-25-00696-f004:**
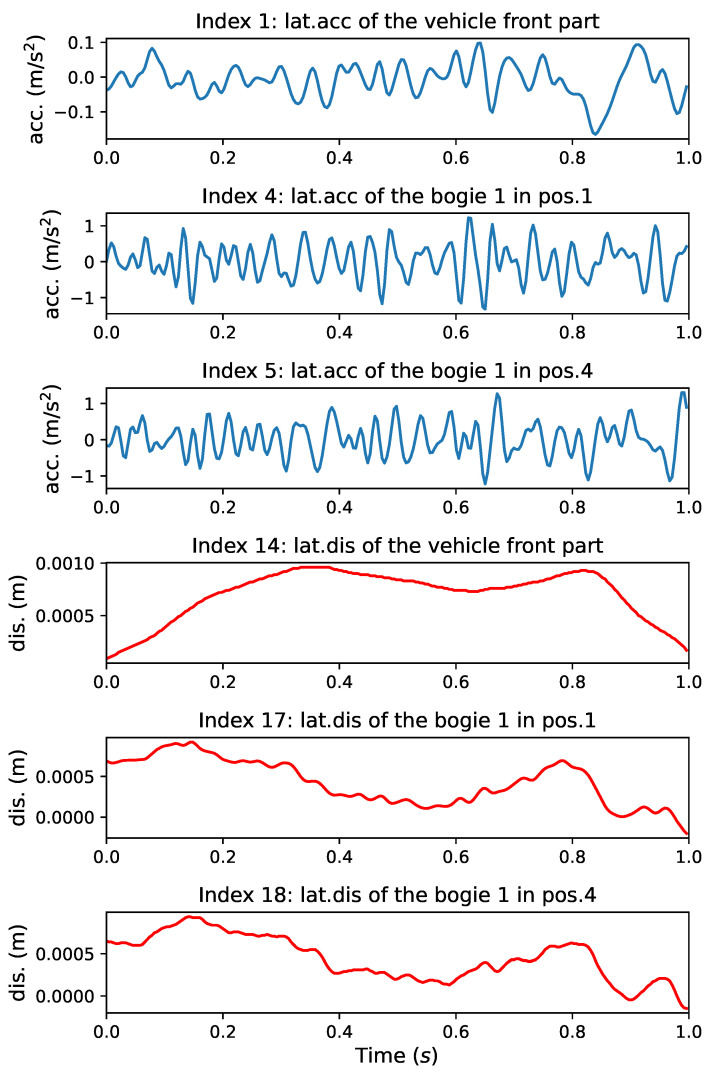
High-speedtrain vibration signal sample.

**Figure 5 entropy-25-00696-f005:**
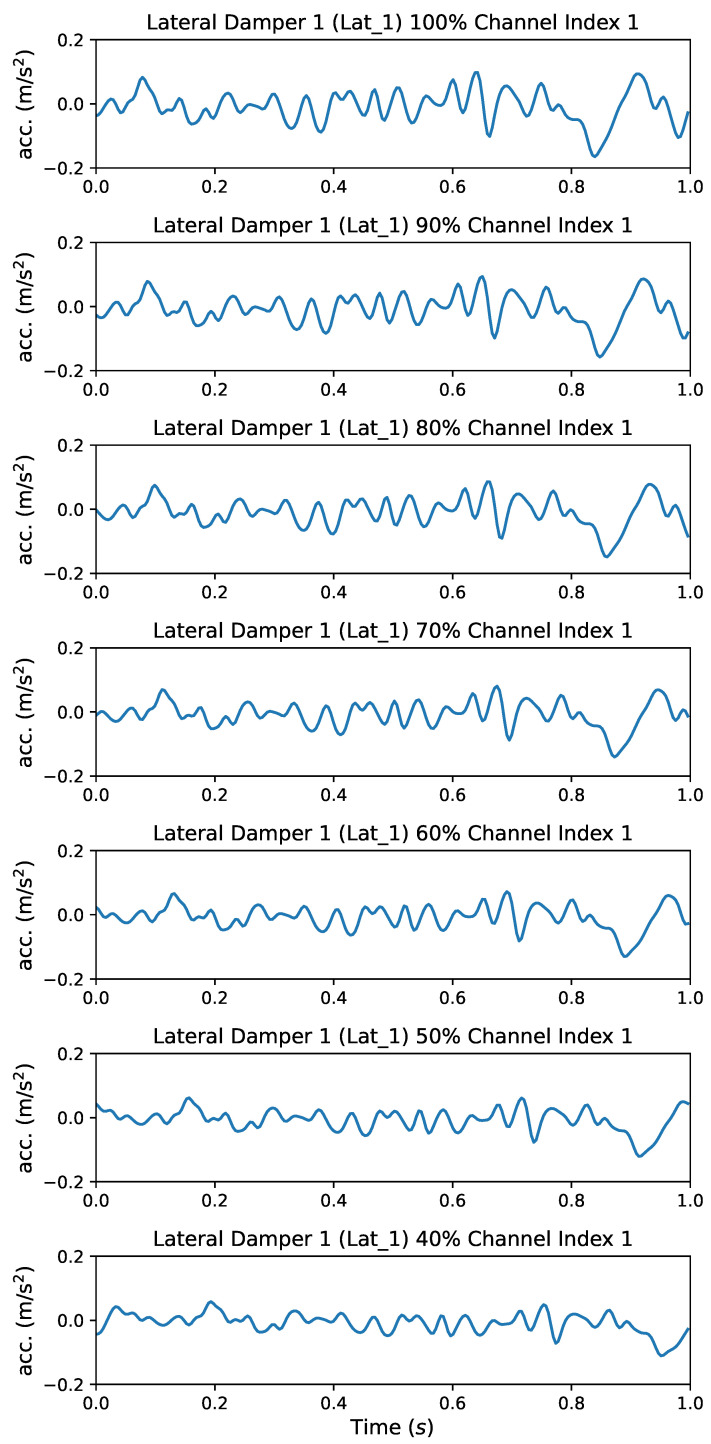
Comparison of vibration signal samples with different states of degradation.

**Figure 6 entropy-25-00696-f006:**
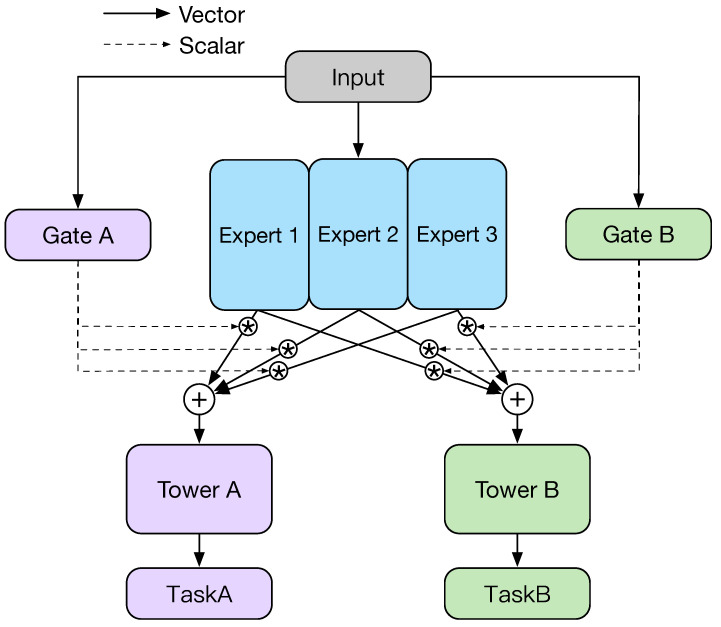
Overall structure of MMoE.

**Figure 7 entropy-25-00696-f007:**
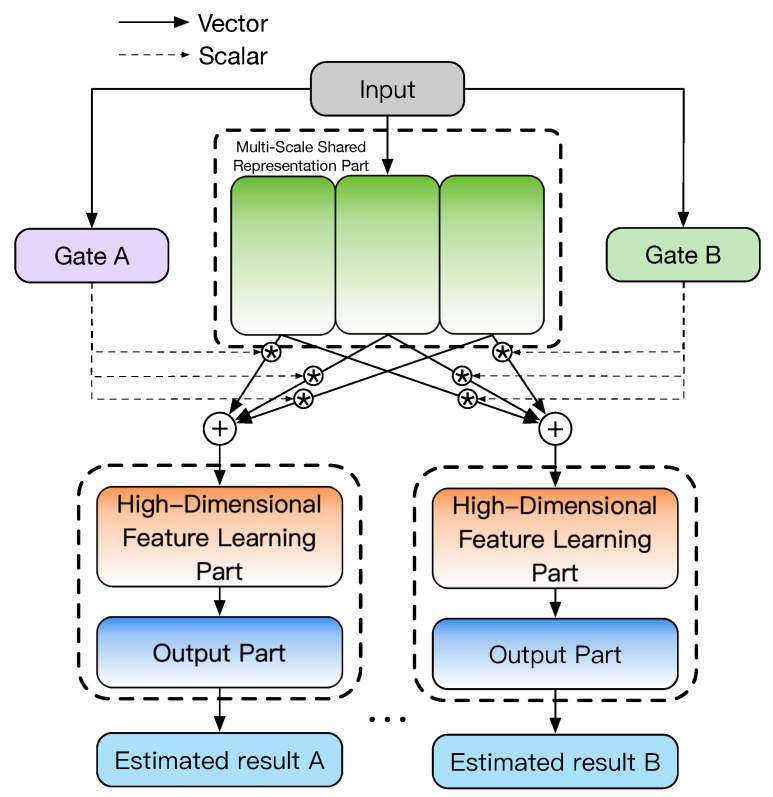
Overall structure of MMCNN.

**Figure 8 entropy-25-00696-f008:**
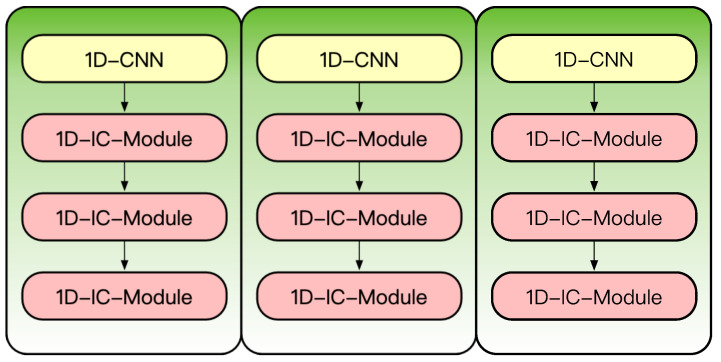
The structure of the multi-scale shared representation part.

**Figure 9 entropy-25-00696-f009:**
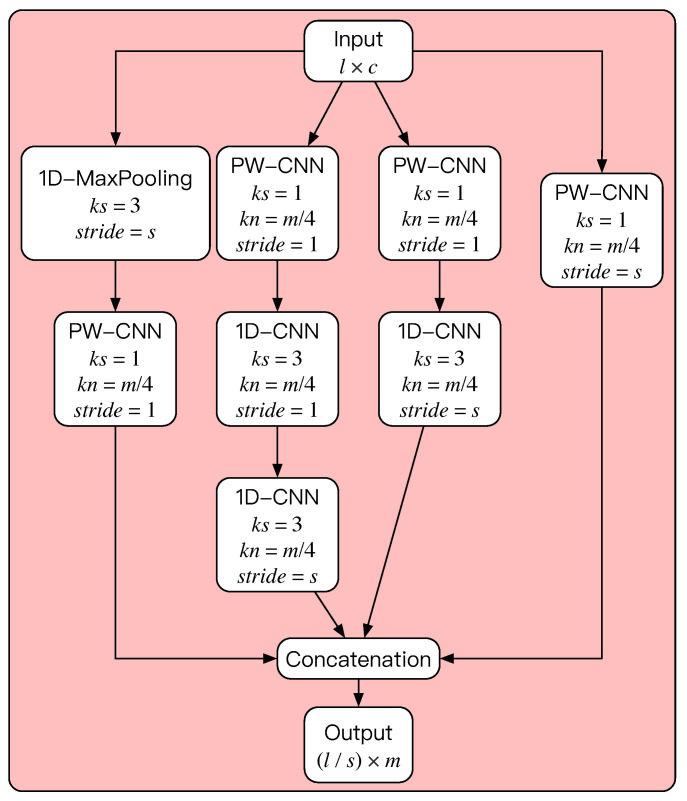
The structure of 1D-IC-Module (in this figure, ks represents the kernel size, kn represents the kernel number, and *s* represents the hyperparameter stride of the whole 1D-IC-Module).

**Figure 10 entropy-25-00696-f010:**
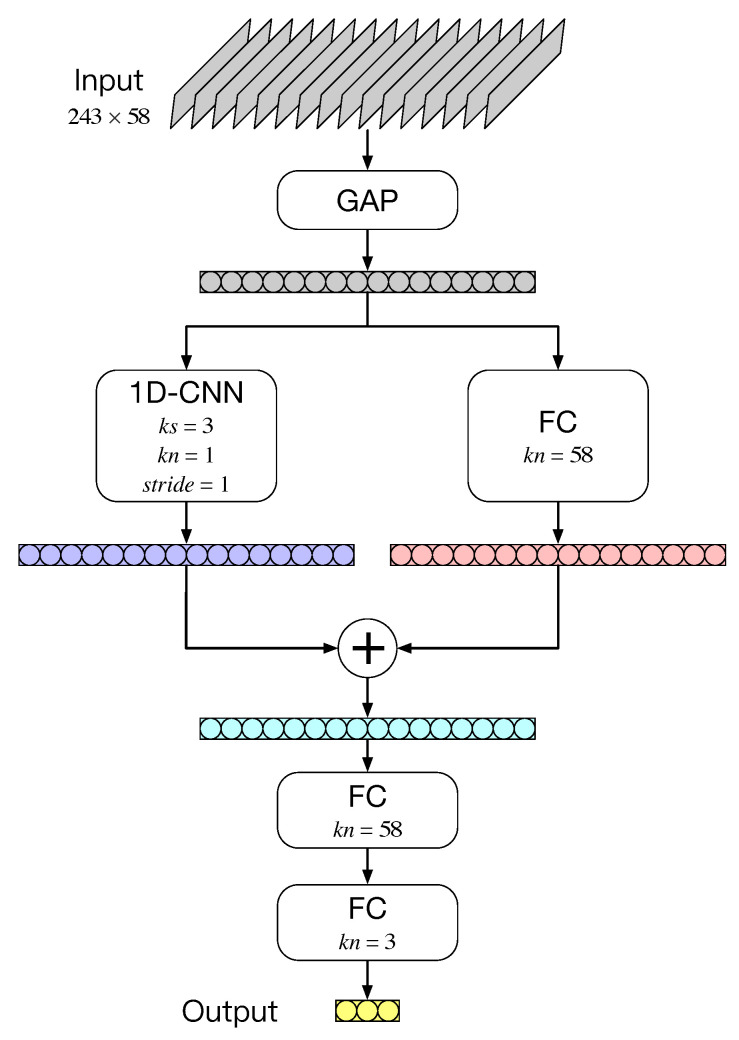
The structure of gate structure (ks represents the kernel size, kn represents the kernel number).

**Figure 11 entropy-25-00696-f011:**
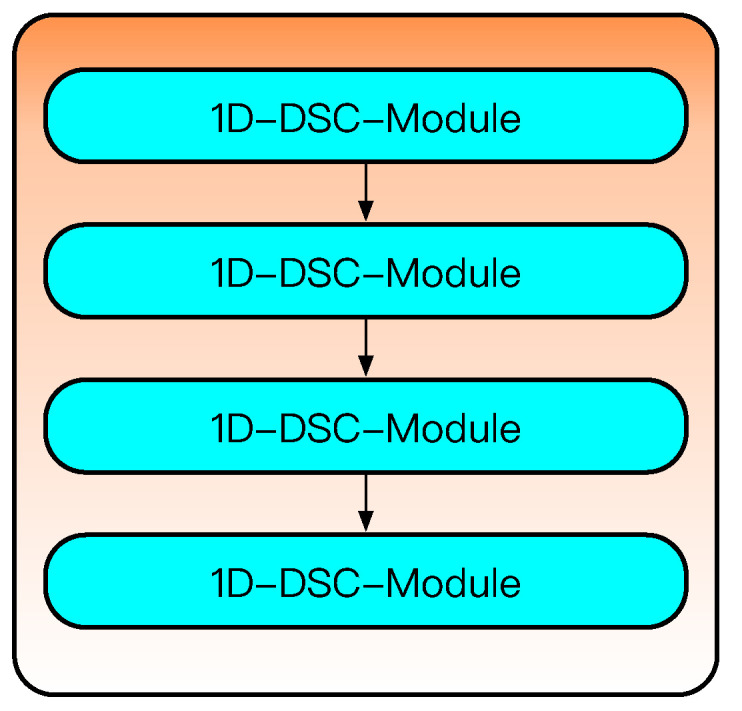
The structure of the high-dimensional feature learning part.

**Figure 12 entropy-25-00696-f012:**
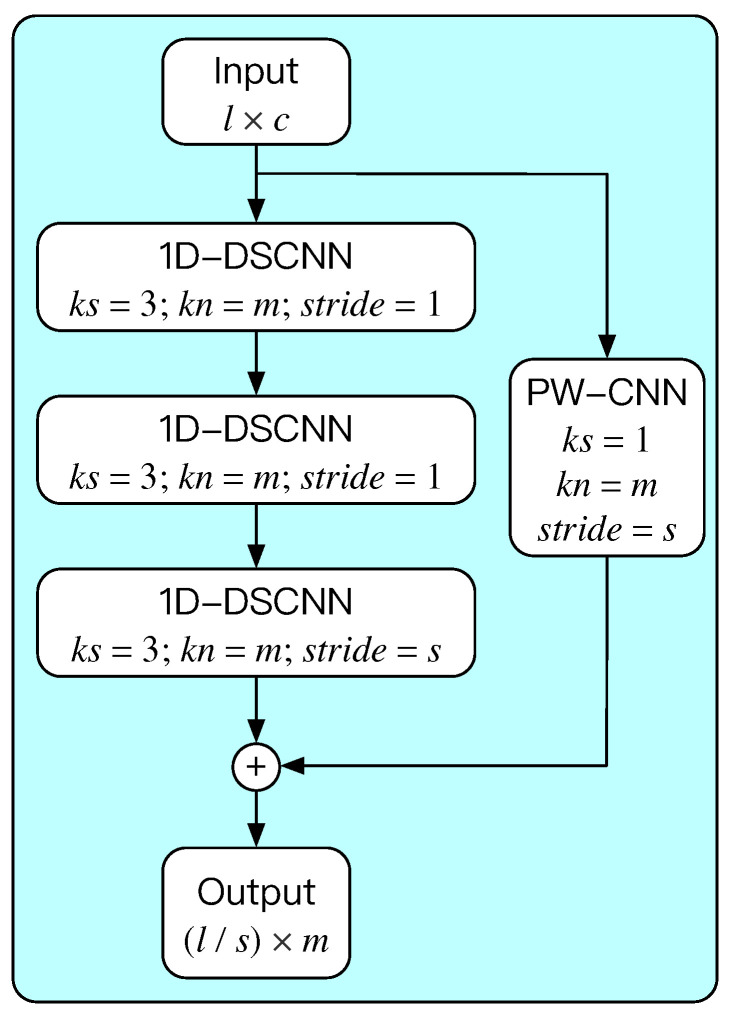
The structure of 1D-DSC-Module (ks represents the kernel size, kn represents the kernel number).

**Figure 13 entropy-25-00696-f013:**
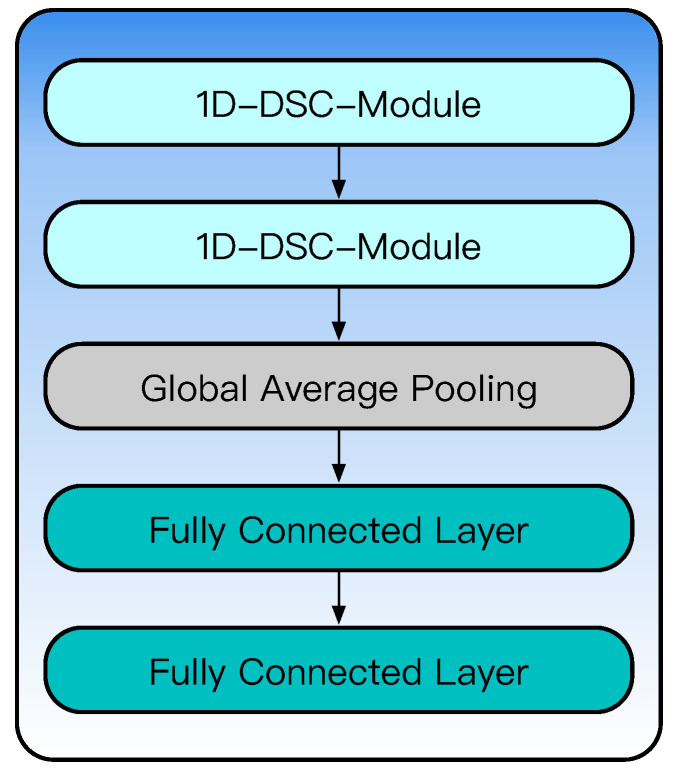
The structure of the output part.

**Figure 14 entropy-25-00696-f014:**
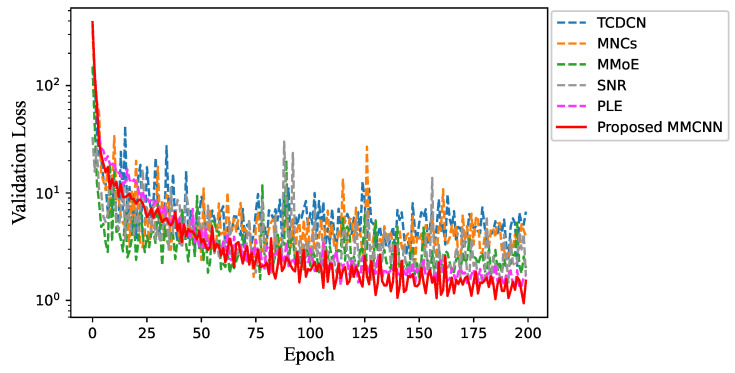
The process of loss decline during training.

**Table 1 entropy-25-00696-t001:** Details of high-speed train signal channels.

Index	Description	Index	Description
1	lat.acc of the veh. front part	30	ver.acc of the axle box 4
2	lat.acc of the veh. mid. part	31	lat.dis of the veh. front part
3	lat.acc of the veh. rear part	32	ver.dis of the veh. front part
4	ver.acc of the veh. mid. part	33	lat.dis of the veh. mid. part
5	ver.acc of the veh. front part	34	ver.dis of the veh. mid. part
6	ver.acc of the veh. rear part	35	lat.dis of the veh. rear part
7	lat.acc of the bog. 1 in pos. 1	36	ver.dis of the veh. rear part
8	ver.acc of the bog. 1 in pos. 1	37	lat.dis of the bog. 1 in pos. 1
9	lat.acc of the bog. 1 in pos. 4	38	ver.dis of the bog. 1 in pos. 1
10	ver.acc of the bog. 1 in pos. 4	39	lat.dis of the bog. 1 in pos. 4
11	lat.acc of the bog. 1 in mid.	40	ver.dis of the bog. 1 in pos. 4
12	ver.acc of the bog. 1 in mid.	41	lat.dis of the bog. 1 in mid.
13	lat.acc of the bog. 2 in pos. 5	42	ver.dis of the bog. 1 in mid.
14	ver.acc of the bog. 2 in pos. 5	43	lat.dis of the bog. 2 in pos. 5
15	lat.acc of the bog. 2 in pos. 8	44	ver.dis of the bog. 2 in pos. 5
16	ver.acc of the bog. 2 in pos. 8	45	lat.dis of the bog. 2 in pos. 8
17	lat.acc of the bog. 2 in mid.	46	ver.dis of the bog. 2 in pos. 8
18	ver.acc of the bog. 2 in mid.	47	lat.dis of the bog. 2 in mid.
19	lon.acc of the axle box 1	48	ver.dis of the bog. 2 in mid.
20	lat.acc of the axle box 1	49	lat.dis of the wheel-set 1
21	ver.acc of the axle box 1	50	lat.dis of the wheel-set 2
22	lon.acc of the axle box 2	51	lat.dis of the wheel-set 3
23	lat.acc of the axle box 2	52	lat.dis of the wheel-set 4
24	ver.acc of the axle box 2	53	rel. dis. of ps. in pos. 1
25	lon.acc of the axle box 3	54	rel. dis. of ps. in pos. 8
26	lat.acc of the axle box 3	55	rel. dis. of ss. in pos. 1
27	ver.acc of the axle box 3	56	rel. dis. of ss. in pos. 8
28	lon.acc of the axle box 4	57	rel. dis. of yaw dam. in pos. 1
29	lat.acc of the axle box 4	58	rel. dis. of yaw dam. in pos. 8

Note: lat. = lateral, ver. = vertical, lon. = longitudinal, acc. = acceleration, dis. = displacement, pos. = position, veh. = vehicle, mid. = middle, bog. = bogie, rel. = relative, ps. = primary suspension, ss. = secondary suspension, dam. = damper.

**Table 2 entropy-25-00696-t002:** The specific hyperparameters of a branch of multi-scale shared representation part.

Layers	Kernel Size	Kernel Number	Stride	Input Size	Output Size
1D-CNN	3	64	2	243 × 58	122 × 64
1D-IC-Module	-	-	2	122 × 64	61 × 128
1D-IC-Module	-	-	2	61 × 128	31 × 256
1D-IC-Module	-	-	2	31 × 256	16 × 728

**Table 3 entropy-25-00696-t003:** The specific hyperparameters of the high-dimensional feature learning part.

Layers	Kernel Size	Kernel Number	Stride	Input Size	Output Size
1D-DSC-Module	-	-	1	16 × 728	16 × 728
1D-DSC-Module	-	-	1	16 × 728	16 × 728
1D-DSC-Module	-	-	1	16 × 728	16 × 728
1D-DSC-Module	-	-	1	16 × 728	16 × 728

**Table 4 entropy-25-00696-t004:** The specific hyperparameters of the output part.

Layers	Kernel Size	Kernel Number	Stride	Input Size	Output Size
1D-DSC-Module	-	-	2	16 × 728	8 × 1024
1D-DSC-Module	-	-	1	8 × 1024	8 × 2048
Global Average Pooling	-	-	-	8 × 2048	1 × 2048
Fully Connected Layer	-	2048	-	1 × 2048	1 × 2048
Fully Connected Layer	-	1	-	1 × 2048	1 × 8

**Table 5 entropy-25-00696-t005:** The training set of mixed performance degradation of all lateral dampers and yaw dampers.

Degradation Damper	Number of Samples	Label
Lat_1	2000	[$x_{i}$,100,100,100,100,100,100,100]
Lat_2	2000	[100,$x_{i}$,100,100,100,100,100,100]
Lat_3	2000	[100,100,$x_{i}$,100,100,100,100,100]
Lat_4	2000	[100,100,100,$x_{i}$,100,100,100,100]
Yaw_1	2000	[100,100,100,100,$x_{i}$,100,100,100]
Yaw_2	2000	[100,100,100,100,100,$x_{i}$,100,100]
Yaw_3	2000	[100,100,100,100,100,100,$x_{i}$,100]
Yaw_4	2000	[100,100,100,100,100,100,100,$x_{i}$]

$x_{i}$ denotes the current degradation damper’s performance degradation state, and there are 7 states in the training set, which are 100%, 90%, ..., 50%, and 40% respectively. There are 2000 training samples for each $x_{i}$.

**Table 6 entropy-25-00696-t006:** The test set of mixed performance degradation of all lateral dampers and yaw dampers.

Degradation Damper	Number of Samples	Label
Lat_1	400	[$x_{i}$,100,100,100,100,100,100,100]
Lat_2	400	[100,$x_{i}$,100,100,100,100,100,100]
Lat_3	400	[100,100,$x_{i}$,100,100,100,100,100]
Lat_4	400	[100,100,100,$x_{i}$,100,100,100,100]
Yaw_1	400	[100,100,100,100,$x_{i}$,100,100,100]
Yaw_2	400	[100,100,100,100,100,$x_{i}$,100,100]
Yaw_3	400	[100,100,100,100,100,100,$x_{i}$,100]
Yaw_4	400	[100,100,100,100,100,100,100,$x_{i}$]

$x_{i}$ denotes the current degradation damper’s performance degradation state, and there are 13 states in the training set, which are 100%, 95%, 90%, ..., 50%, 45%, and 40% respectively. There are 400 test samples for each $x_{i}$.

**Table 7 entropy-25-00696-t007:** The training set of lateral dampers.

Degradation Damper	Number of Samples	Label
Lat_1	2000	[$x_{i}$,100,100,100]
Lat_2	2000	[100,$x_{i}$,100,100]
Lat_3	2000	[100,100,$x_{i}$,100]
Lat_4	2000	[100,100,100,$x_{i}$]

$x_{i}$ denotes the current degradation damper’s performance degradation state, and there are 7 states in the training set, which are 100%, 90%, ..., 50%, and 40% respectively. There are 2000 training samples for each $x_{i}$.

**Table 8 entropy-25-00696-t008:** The test set of lateral dampers.

Degradation Damper	Number of Samples	Label
Lat_1	400	[$x_{i}$,100,100,100]
Lat_2	400	[100,$x_{i}$,100,100]
Lat_3	400	[100,100,$x_{i}$,100]
Lat_4	400	[100,100,100,$x_{i}$]

$x_{i}$ denotes the current degradation damper’s performance degradation state, and there are 13 states in the training set, which are 100%, 95%, 90%, ..., 50%, 45%, and 40% respectively. There are 400 test samples for each $x_{i}$.

**Table 9 entropy-25-00696-t009:** The training set of yaw dampers.

Degradation Damper	Number of Samples	Label
Yaw_1	2000	[$x_{i}$,100,100,100]
Yaw_2	2000	[100,$x_{i}$,100,100]
Yaw_3	2000	[100,100,$x_{i}$,100]
Yaw_4	2000	[100,100,100,$x_{i}$]

$x_{i}$ denotes the current degradation damper’s performance degradation state, and there are 7 states in the training set, which are 100%, 90%, ..., 50%, and 40% respectively. There are 2000 training samples for each $x_{i}$.

**Table 10 entropy-25-00696-t010:** The test set of yaw dampers.

Degradation Damper	Number of Samples	Label
Yaw_1	400	[$x_{i}$,100,100,100]
Yaw_2	400	[100,$x_{i}$,100,100]
Yaw_3	400	[100,100,$x_{i}$,100]
Yaw_4	400	[100,100,100,$x_{i}$]

$x_{i}$ denotes the current degradation damper’s performance degradation state, and there are 13 states in the training set, which are 100%, 95%, 90%, ..., 50%, 45%, and 40% respectively. There are 400 test samples for each $x_{i}$.

**Table 11 entropy-25-00696-t011:** Ablation experiment results of multi-scale shared representation part.

Module	MAE	RMSE
Inception V1	2.31	2.94
Inception-ResNet	2.02	2.36
ResNeXt	1.91	2.21
Res2Net	1.68	1.93
1D-MSIC-Module	**1.28 **	**1.56**

**Table 12 entropy-25-00696-t012:** Ablation experiment results of high-dimensional feature learning part.

Module	MAE	RMSE
Inception V1	1.97	2.45
Inception-ResNet	1.54	1.77
ResNeXt	1.61	1.90
Res2Net	1.43	1.72
1D-MSDSC-Module	**1.28**	**1.56**

**Table 13 entropy-25-00696-t013:** Ablation experiment results of different numbers of 1D-IC Modules in the multi-scale shared presentation part.

Number	MAE	RMSE
2	1.82	2.21
3	**1.28**	**1.56**
4	1.37	1.72
5	1.52	1.83

**Table 14 entropy-25-00696-t014:** Ablation experiment results of different numbers of 1D-IC Modules in the high-dimensional feature learning part.

Number	MAE	RMSE
2	1.52	1.87
3	1.39	1.68
4	**1.28**	**1.56**
5	1.34	1.62

**Table 15 entropy-25-00696-t015:** Ablation experiment results of gate structure.

Attentional Mechanisms	MAE	RMSE
SE-Net	1.58	1.89
Proposed gate structure	**1.28**	**1.56**

**Table 16 entropy-25-00696-t016:** Results of multi-task range experiment.

Multi-Task Range	MAE	RMSE
All dampers (Table 5 with Table 6)	1.28	1.56
Only lateral dampers (Table 7 with Table 8)	**0.78**	**0.95**
Only yaw dampers (Table 9 with Table 10)	1.02	1.16

**Table 17 entropy-25-00696-t017:** Results of comparison experiment.

MTL Type	Models	MAE	RMSE
Hard parameter sharing	TCDCN	3.74	4.68
	MNCs	3.13	3.89
Soft parameter sharing	MMoE	2.13	2.72
	SNR	1.87	2.33
	PLE	1.57	1.84
	Proposed MMCNN	**1.28**	**1.56**

MTL represents multi-tasking learning.

## Data Availability

Not applicable.

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
