# Peer review of "Research on Performance Degradation Estimation of Key Components of High-Speed Train Bogie Based on Multi-Task Learning"

_entropy, 2023, doi:10.3390/e25040696_

Round 1
Reviewer 1 Report
* A brief summary:
The article proposes an architecture to perform degradation estimation in several components of a high-speed train bogie.
The architecture elaborates on previous versions and propose a multi task and multi-scale convolutional neural network, MMCNN, capable to estimate accurately the degradation in different parts of the bogie, for instance, in different types of dampers. Multi-scale feature learning is improved using a multiscale one-dimensional convolution models, instead of an ordinary convolutional layer, which is the basis for MMoE (the basic architecture on which this work elaborates).The proposal also includes soft parameter sharing method because several parameters are common to different parts of the bogie.
The work is tested on simulated data on a particular vehicle from a widely recognized test rig at the Southwest Jiaotong University.
Results show the good results of the approach against other methods and architectures.
* General concept comments:
The article addresses a complex problem, providing a solution which works well in simulation.
Regarding the experimental dataset description, just based on figure 4 is very difficult to imagine the complexity of the degradation estimation.
Providing some additional graphs where the effects of degradation can be seen along the 2000 samples would help to understand the complexity of the problem at hand.
A more precise description of the proposed architectures (regarding number of inputs, neurons in each layer, etc.) would help in reproducing the proposal.
Authors claim in the conclusions that the proposed architecture reduce calculation costs, but provide no evidence in this matter. This should be included or the claim removed.
There is also no discussion along the paper about the relation between the proposed architecture (which include 3 sequential 1D-IC-Modules x 3 in the multiscale part or 4 x 1D-DSC-Moudles,
or 3 x 1D-DSCNN layers in the 1D-DSC-Module) and the different tasks or the different scales in the problem at hand. This would help to understand how to apply the solution to other domains.
* Specific comments:
References to different NN architectures in the bibliography must be in capital letters, such as 1D-ConvLSTM, PLE or other terms such as MTL or journals/conferences such as ECCV or CVPR are provided without explaining the complete reference.
There are multiple typos in the article, being the most important two missed references within the text:
- pg. 4, line 163, missing "ref Description"
- pg. 13, linea 258, missing ref "in XXX"
Additionally:
pg. 1, line 36 : means -> mean; line 66: "lateral damper" --> "lateral dampers"; line 68 "grouped and input" --> "groups as input??"
pg. 4. line 123: from "train" in --> trains; same for vehicle(s) in line 124; line 128: "as which" --> which
pg. 13, line 254, references to RMSE(1) and MAE(2) must be (2) and (1) or change the order for formulas (1) and (2).
Author Response
Point 1: Regarding the experimental dataset description, just based on figure 4 is very difficult to imagine the complexity of the degradation estimation. Providing some additional graphs where the effects of degradation can be seen along the 2000 samples would help to understand the complexity of the problem at hand.
Response 1: We appreciate the reviewer’s valuable suggestions. In order to futher explain the complexity of the problem, we have added a figure to compare the differences among signal samples under different performance degradation states. To make it clear, the following revisions are added in the manuscript in appropriate places:
(2. Data Description, paragraph 2) Figure 5 shows the vibration signal samples of high-speed train lateral damper (Lat_1) in different performance degradation states at the same driving position and time interval. The vibration modes of these samples are very close, and it is not easy to accurately estimate their performance degradation.
Point 2: A more precise description of the proposed architectures (regarding number of inputs, neurons in each layer, etc.) would help in reproducing the proposal.
Response 2: We have modified some figures of the proposed achitectures and added some tables to describe the architectures more accurately and help in reproducing the proposal. To make it clear, the following revisions are added in the manuscript in appropriate places:
(3.1 Multi-Scale Shared Representation Part)
(3.2 Gate Structure)
(3.3 High-Dimensional Feature Learning Part)
(3.4 Output Part)
Point 3: Authors claim in the conclusions that the proposed architecture reduce calculation costs, but provide no evidence in this matter. This should be included or the claim removed.
Response 3: Thank you for pointing out this issue. The description of "reduce calculation costs" has been deleted, and to make it clear, the following revisions are added in the manuscript in appropriate places:
(5. Conclusion) Compared with the models in the existing performance degradation studies, the proposed model can estimate the performance degradation of multiple dampers at the same time.
Point 4: There is also no discussion along the paper about the relation between the proposed architecture (which include 3 sequential 1D-IC-Modules x 3 in the multiscale part or 4 x 1D-DSC-Moudles, or 3 x 1D-DSCNN layers in the 1D-DSC-Module) and the different tasks or the different scales in the problem at hand. This would help to understand how to apply the solution to other domains.
Response 4: We have increased two experiments in the ablation experiment part to expain the impacts on experimental results of different amount of 1D-IC-Modules and 1D-DSC-Moudles adopted. 3 x 1D-DSCNN layers in the 1D-DSC-Module has refered to existing research results of high-speed train bogie performance degradation estimation. When the solution of this paper is applied in other domains, the number of different modules adopted in this architecture can be adjusted through experiments. The following revisions are added in the manuscript in appropriate places:
(4.1 Ablation Experiments) The ablation experiment also discussed the impacts on performance degradation estimation results while adopting different amounts of 1D-IC Modules and 1D-DSC-Modules. Firstly, the experiment compared the situations of using different quantities of 1D-IC Modules in the multi-scale shared presentation part, and the experimental results are shown in Table 13. The experimental results indicate that the minimum estimation error can be achieved when three 1D-IC modules are applied in the multi-scale shared representation part. Secondly, the experiment compared the situations of adopting different numbers of 1D-DSC-Modules in the high dimensional feature learning part, and the experimental results are shown in Table 14. The experimental results indicate that the minimum estimation error can be achieved when four 1D-DSC-Modules are applied in the high dimensional feature learning part. The experimental results also demonstrate that adopting different amounts of multi-scale modules can have significant impacts on the estimation results. When the proposed structure is applied to performance degradation estimation tasks in other fields, the number of multi-scale modules adopted needs to be adjusted based on actual experimental results.
Point 5: References to different NN architectures in the bibliography must be in capital letters, such as 1D-ConvLSTM, PLE or other terms such as MTL or journals/conferences such as ECCV or CVPR are provided without explaining the complete reference.
Response 5: We have capitalized the references to different NN architectures in the bibliography and explained the complete reference.
Point 6: There are multiple typos in the article, being the most important two missed references within the text:
- pg. 4, line 163, missing "ref Description"
- pg. 13, linea 258, missing ref "in XXX".
Response 6: We have added the missed references.
Point 7: pg. 1, line 36 : means -> mean; line 66: "lateral damper" --> "lateral dampers"; line 68 "grouped and input" --> "groups as input??"
Response 7: We have corrected the usage of words, expressions, and grammar points.
Point 8: pg. 4. line 123: from "train" in --> trains; same for vehicle(s) in line 124; line 128: "as which" --> which.
Response 8: We have corrected the usage of words, expressions, and grammar points.
Point 9: pg. 13, line 254, references to RMSE(1) and MAE(2) must be (2) and (1) or change the order for formulas (1) and (2).
Response 9: We have revised the reference numbers.

Reviewer 2 Report
An interesting work and results.
I appreciate the experimental facility used to generate the physical signals for fault detection.
A model of fault diagnosis should be considered soon or later.
The manuscript has an abstract which presents the problem, the status of the current research and the contributions of the proposed paper.
The basic point is to consider the problem of fault detection at the level of the system, and not at the level of components.
As far I understood, the topic is relevant for the fault detection problems and for High-Speed Train Bogies.
The paper proposes a multi task and a multi-scale convolutional neural network approach.
Conclusions are consistent with the evidence and arguments presented.
The list of references is appropriate.
Author Response
Thank you very much for your review and affirmation.